

# A functional approach to the body condition assessment of lactating donkeys as a tool for welfare evaluation

Emanuela Valle[1], Federica Raspa[1], Marzia Giribaldi[2], Raffaella Barbero[3], Stefania Bergagna[3], Sara Antoniazzi[2], Amy K. Mc Lean[4], Michela Minero[5] and Laura Cavallarin[2]

[1] Department of Veterinary Science, University of Turin, Grugliasco (Torino), Italy
[2] ISPA-CNR, Institute of Sciences of Food Production, Grugliasco, Italy
[3] The Veterinary Medical Research Institute for Piemonte, Liguria and Valle d'Aosta, Torino, Italy
[4] North Carolina State University, Animal Science, Raleigh, NC, USA
[5] Department Veterinary Science and Public Health, University of Milano, Milano, Italy

Corresponding author
Emanuela Valle,
emanuela.valle@unito.it

## ABSTRACT

**Background**. The breeding of lactating donkeys is increasing in Western Europe; with it the evaluation of body condition is growing in importance since it is considered a key principle for their welfare. However, assessment of body condition is a complex task, since several factors are involved. The aim of the present study is to investigate which animal-based indicators are the most reliable to describe the body condition of lactating donkeys. For this purpose, new animal-based indicators, which are easy to measure in field conditions (including body measurements, fatty neck score (FNS), dental score), are recorded and their relationship with BCS (a proxy measure for overall adiposity) was assessed. The ones that reveal an association with the BCS are included in an integrated principal component analysis to understand which are the most related to BCS.

**Methods**. Fifty-three healthy lactating donkeys of various breeds, including 7 Martina Franca, 10 Ragusano, 2 Romagnolo and 34 crossbreeds, were evaluated. The animal-based indicators that were recorded were: length (OP, olecranon tuber-pinbone and SH, shoulder-hip), heart girth (HG), abdominal circumference (AC), neck length (NL), neck height (NH) and neck thickness (NT) at 0.50 and neck circumference (NC) at 0.25, 0.50 and 0.75, body condition score (BCS) and fatty neck score (FNS). The owners' evaluation of the BCS was also considered. A dental assessment was performed and the month of lactation and age of each animal was recorded.

**Results**. No correlation was found between BCS and the other morphometric body measurements. On the contrary the FNS was correlated with the morphometric measurements of the neck (positive correlation to 0.50 NH and 0.50 NT, 0.50 NC, 0.75 mean NC, and negative correlation to the mean NC:NH and mean NC:NT, 0.50 NC:NT and 0.50 NC:NH ratios). A significant inverse relationship was identified between BCS and dental score. A Principal Component analysis (PCA) separated the BCS classes on the first principal component (PC1). PC1 revealed a meaningful positive correlation between the BCS and the neck measurements (NT, NH and FNS), with high positive loadings, while a negative correlation was found for dental abnormalities. The owners' evaluation of BCS was different from the expert evaluator' assessment, since they tended to give higher score that was slightly but significantly correlated to AC.

**Discussion**. A new scoring system, called Fatty Neck Score (FNS), has been proposed for the judgement of the adiposity status of donkey neck. The results suggest that caregivers might use the proposed animal based indicators (BCS, FNS and dental scores) together as a tool for the evaluation of the body condition of lactating donkeys. Our findings highlight that caregivers need to be trained in order to be able to properly record these indicators. Ultimately use of these indicators may help to improve the welfare of lactating donkeys.

## INTRODUCTION

Donkey's milk has long been used as a substitute for human milk, and its use is related to the countries in which donkeys are traditionally bred, such as Asia, Africa and Eastern Europe (*Vincenzetti, Polidori & Vita, 2007*). However, in the last 10 years, the breeding of donkeys for the production of milk has increased, as has the number of published research papers on the use of donkey milk, especially in Western Europe (http://www.gopubmed.org/). This is due to the fact that donkey milk has been demonstrated to be a useful substitute for children who are affected by allergies to the milk proteins of cows, or who suffer from multiple food intolerances (*Monti et al., 2007*; *Monti et al., 2012*). It has recently been demonstrated *in vivo* that dietary supplementation with donkey milk can result in a decrease in the inflammatory status of Wistar rats used as an animal model, and that this decrease is in turn associated with an improvement in the lipid and glucose metabolism, compared to a diet supplemented with bovine milk (*Trinchese et al., 2015*). Donkey milk also has a long tradition of cosmetic use (*Faye & Konuspayeva, 2012*). These potential uses of donkey milk have led to an increase in the donkey population in Western Europe, and in particular, to the number of animals that are bred in Italy (*D'Alessandro & Martemucci, 2012*; *Cavallarin et al., 2015*).

Since the breeding of lactating donkeys is increasing, it is important to evaluate their welfare. Body condition can be considered a key criterion of the overall welfare of the animals. Even though the body condition has been studied extensively, the term includes a series of concepts that are not clearly expressed in many situations. Generally speaking, the "condition" of an animal is considered as an indicator of its "individual quality" (*Labocha, Schutz & Hayes, 2013*). In most studies, the body condition is treated as a measure of energy reserves (mostly fat mass), but it is difficult to measure directly; in fact, direct methods, such as dilution techniques or dual-energy X-ray absorptiometry (*Quaresma, Payan-Carreira & Silva, 2013*), suffer from noteworthy limitations, since they are expensive and impractical to perform on a farm.

For this reason, the body condition is inferred from several animal-based indicators that are proposed to describe energy, reserves or mass of body fat; some of these indicators are morphometric, while others are biochemical or physiological measurements (*Labocha & Hayes, 2012*).

One of the most frequently used animal-based morphometric indicator is body condition score (BCS). This is a proxy measure of the total mass of a subcutaneous fat, know as adiposity of the body. By means of a visual appraisal and palpation of the adipose tissue sites, it is possible to rate the animal's body condition using a numerical scale (*Carter & Dugdale, 2013*). Different BCS scoring systems, based on 5- or 9-point scales, are also available for donkeys (*Pearson & Ouassat, 2000*; *Burden, 2012*). Other animal-based morphometric indicators, such as direct measurements of the body or calculation of the ratio index, can be used, and could theoretically reduce the bias of subjective scoring (as in the case of BCS) if used simultaneously (*Carter & Dugdale, 2013*). The association between these animal-based morphometric indicators and BCS, has been evaluated in different studies on ponies and horses (*Carter et al., 2009*; *Dugdale et al., 2011*; *Fernandes et al., 2015*; *Martinson et al., 2014*; *Giles et al., 2014*), and donkeys (*Cappai, Picciau & Pinna, 2013*) but knowledge is still scant where donkeys are concerned.

Moreover, other animal-based morphometric indicators have been proposed in the last few years; many studies have identified that equids, store regional fat especially on the neck, that can remain in donkeys even if the overall condition decreases (*Burden, 2012*; *Burden & Thiemann, 2015*). A proposal for a neck score of the adiposity of donkeys exists (*Mendoza et al., 2015*), but is based on a 0–4 scale, rather than on the 0–5-point scale and it is not associated with all the morphometric variables of the neck, such as the neck circumference (NC) measured at 0.25 and 0.75 of the neck length, the neck height (NH) or the neck thickness (NT) at 0.50. We therefore propose a new measurement which we refer to as the 'Fatty Neck Score'

In addition to the animal-based morphometric indicators used to describe the body condition, other parameters should also be considered. Both internal and external features affect the body condition of animals and the potential influence of these features should be considered in order to understand the complexity of this concept (*Resano-Mayor et al., 2016*). Age, physiological status and dental health status (*Du Toit et al., 2008*; *Du Toit, Burden & Dixon, 2009*) can be considered as animal-based indicators that can influence the body condition. Moreover, because the owners play an important role in maintaining the body condition, it is also important to assess their perception of the body condition in terms of BCS (*Hemsworth & Coleman, 2011*).

The aim of the study was to identify which morphometric measurements are most practical to describe the body conditions of lactating donkey.

To achieve this goal, we assessed the relationship between body condition score, body measurements, and we tested new measurements, namely the fatty neck score, and a simple dental score. A principal component analysis was conducted on the animal-based indicators that showed an association with BCS, in order to determine which were most useful in defining the body condition of lactating donkey.

## MATERIALS AND METHODS

This paper describes the results of a surveillance program that was set up by the regional health authorities (regional surveillance working group, Protocol 9641/DB2017, 23/03/12).

All the study procedures, none of which involved invasive experimental work, were conducted in the presence of the regional veterinary services. The group was working on the production of regional guidelines on donkey milk production (Regione Piemonte, BU 29 18/07/2013 Code DB2017 D.D. 17 June 2013, no. 461).

## Population description

Monitoring was conducted from May to June 2014 on dairy farms in the North-West of Italy. The study included all the animals present throughout the territory that were milked for the sale of milk. The animals were housed on 6 breeding farms authorized to produce and sell donkey milk, according to the Piedmont Region guidelines (Code DB2017 D.D. 17 June, 2013, no. 461). All the donkeys included in the study were bred on semi-extensive/extensive farms, and had free access to drinking water and forage.

## Animal-based morphometric indicators

### Morphometric indicators of the body

The following body measurements were assessed for each animal, using a soft measuring tape (see Figs. 1 and 2): (i) body length (OP, olecranon tuber-pinbone), measured from the back of the shoulder (olecranon tuber) to the pin bone (ischiatic tuberosity); (ii) body length (SH, shoulder-hip), measured from the shoulder point (intermediate tubercle of the humerus) to the hip (tuber coxae); (iii) heart girth (HG), measured as the circumference of the body, at the point caudal to the elbow (olecranon tuber) 2 cm behind the highest point of the withers; (iv) abdominal circumference (AC), measured at two-thirds of the distance from the shoulder point to the hip. The body weight (BW) was calculated using the formula proposed by *Pearson & Ouassat (2000)*: BW (kg) $= [HG \ (cm)^{2.12} \times OP \ (cm)^{0.688}]/3801$.

Four independent expert evaluators (researchers), rated the body condition score (BCS) on a scale of 1 (poor) to 5 (obese), using a previously established scoring system (*Burden, 2012*). The median of the 4 scores, rounded to the nearest whole or half-score increment, was used for the analysis. The owner was also asked to evaluate the BCS according to the same 5-point scale proposed by *Burden (2012)*. Both the expert evaluators and the owners scored the animals with the help of a chart in which the different scores were defined, and rated the BCS after palpation and a visual assessment of the animals.

### Morphometric indicators of the neck

The following neck measurements were conducted for of each animal, using a soft measuring tape: (i) neck circumference (NC), measured at 0.25, 0.50 and 0.75 of the total neck length; (ii) neck height (NH), measured at 0.50 of the neck length, taken from the dorsal midline of the neck to the point of the estimated differentiation between the crest (tissue apparent above the *ligamentum nuchae*) and the neck musculature; (iii) neck thickness (NT) was introduced considering anecdotal evidence that the neck of a donkey tends to drop sideways to the crest of the neck measured from one side of the neck to the other at 0.50 of the neck length, taken from the point of the estimated differentiation between the crest and the neck musculature; (iv) neck length (NL), measured from the poll to the highest point of the withers.

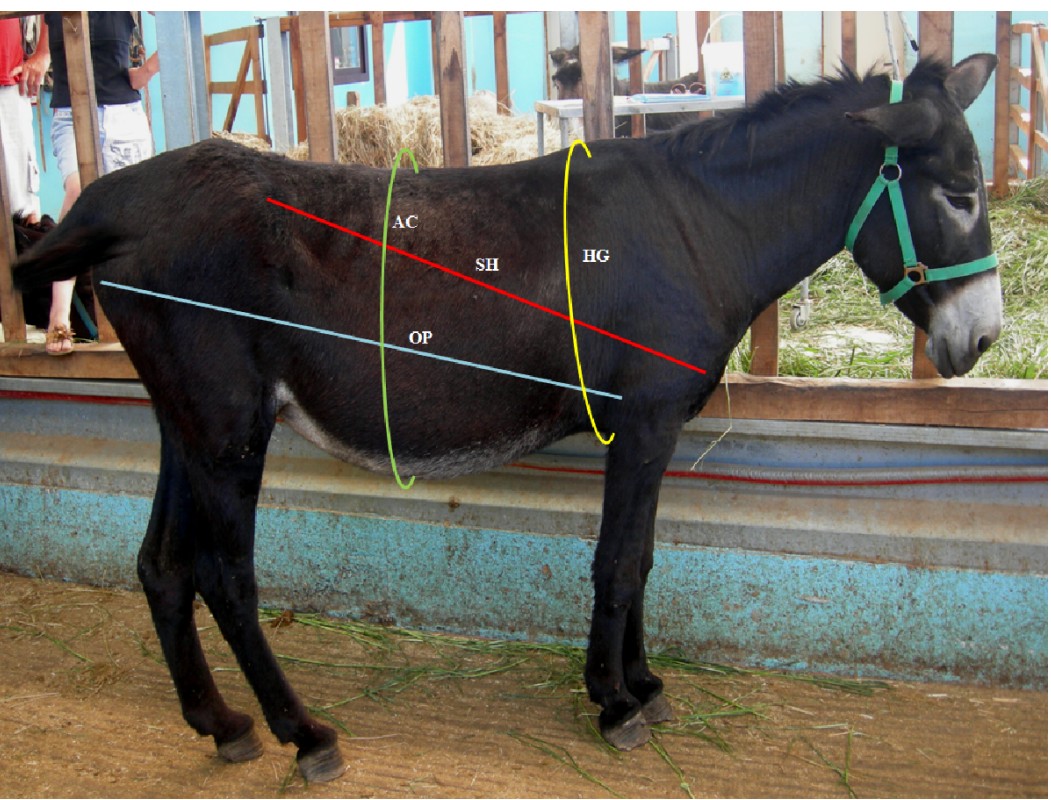

**Figure 1** **Morphometric measurement of the body.** Blue line: body length from the olecranon tuber to the pin bone (OP); Red line: body length from the shoulder point to that of the hip (SH); Yellow line: heart girth (HG), circumference of the body at the point caudal to the elbow, 2 cm behind the highest point of the withers; Green line: abdominal circumference (AC) at two-thirds of the distance from the shoulder point to that of the hip.

After the evaluation of the BCS, the expert evaluators judged the neck fat deposition. They used the fatty neck score (FNS), developed considering the 0–5 point scale that is reported in Table 1, which was based on the one that had previously been proposed for horses (*Carter et al., 2009*). The median of the 4 scores, rounded to the nearest whole or half-score increment, was used for the analysis.

### Dental score assessment

The dental condition of the donkeys was measured on a scale of 0–2, by the same group of expert evaluators. Since it is somewhat difficult to open and look inside a donkey's mouth, the scale was based on simple evaluations that included the appearance of the incisors, palpation of the cheek teeth to identify dental abnormalities, such as sharp points and hooks, quidding and inability to chew. A scale of 0 was used to indicate "normal dental conditions" (good incisors, no sharp points-hooks, no quidding), 1 was used to indicate "discrete dental conditions" (the presence of sharp points or hooks, but still good chewing ability) and 2 was adopted to indicate "poor dental conditions" (the presence of damaged incisors and/or the presence of sharp points and hooks with quidding).

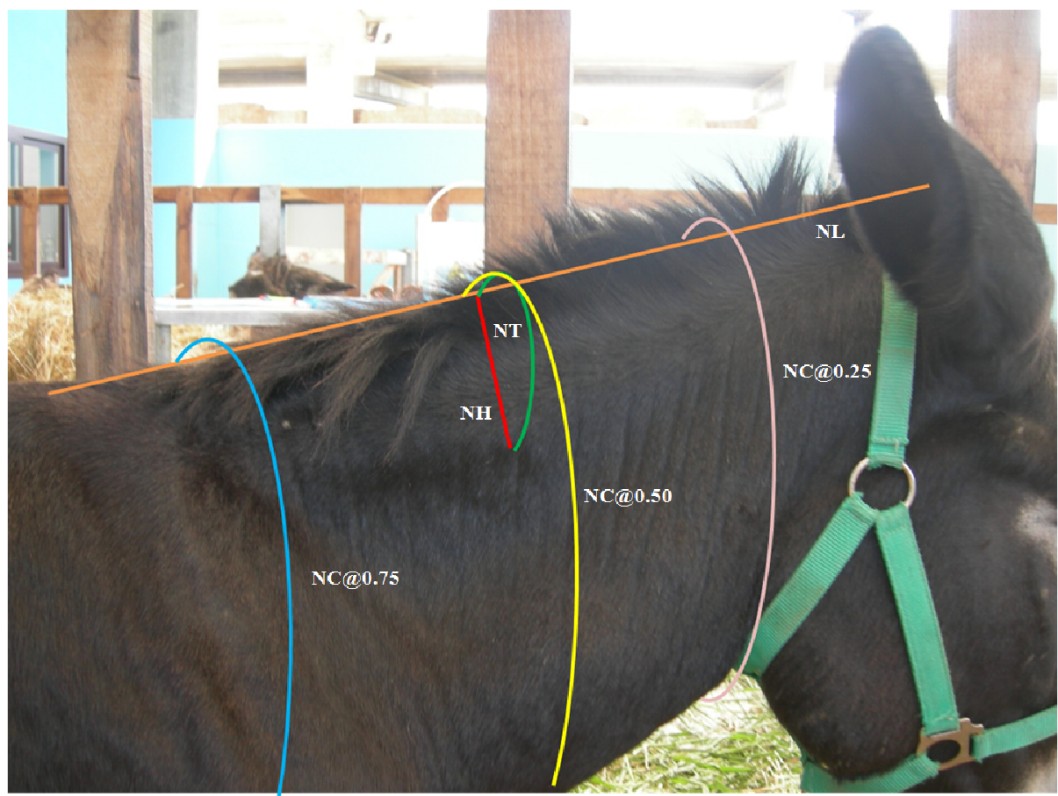

**Figure 2  Morphometric measurement of the neck.** Orange line: neck length (NL), from the poll to the highest point of the withers; Pink, yellow and blue lines: neck circumference (NC), at 0.25, 0.50 and 0.75 of the neck length; Red line: neck height (NH), at 0.50 of the neck length, taken from the dorsal midline of the neck to the point of the estimated differentiation between the crest and the neck musculature; Green line: neck thickness (NT), from one side of the neck to the other at 0.50 of the neck length, taken from the point of the estimated differentiation between the crest and the neck musculature

## Statistical analyses

The statistical analyses were performed with the IBM SPSS Statistics 21 software (SPSS Inc., Chicago, USA) and PAST (version 3.14). The median (interquartiles) and mean values (± standard deviation, SD) were calculated for the following parameters: age, month of lactation and body measurements.

The following indices were calculated: mean NC (average of 0.25 NC, 0.50 NC, 0.75 NC), 0.50NC:NL, mean NC:NL, 0.50NC:NH, Mean NC:NH, 0.50NC:NT, mean NC:NT, AC:BW, AC:HG, AC:OP, AC:SH, HG:BW, HG:OP and HG: SH. The possible associations between the variables were quantified using Spearman's rank correlation coefficient ($r_s$), and the Bonferroni correction was applied to adjust the $p$ values for multiple comparisons. A principal component analysis (PCA) (correlation matrix) was applied to reduce the variables to factors; data assumption for multivariate normality was checked by means of Keiser-Meyer-Olkin (KMO) and Barlett tests, which were performed to test the suitability of the data for structure detection. Only factors with higher Eigenvalues than 1 were considered. Only animal-based indicators that had a significant correlation with BCS, according to $r_s$, were included in the PCA. Among them only the variables that were easy

**Table 1  Fatty neck score (FNS) for donkeys.** Donkey neck images were drawn by Federica Raspa.

| Score | Illustrations of the individual fatty neck score | Description | Neck thickness range according to FNS (in cm) |
|---|---|---|---|
| 0 |  | Neck: thin with the absence of a visible and palpable crest. | <14 |
| 1 |  | Neck: thin with no visible crest, but a slight filling felt upon palpation. | >14–19 |
| 2 |  | Neck: with a moderate deposition of fat. Noticeable appearance of a crest, with fat deposited fairly evenly from the poll to the withers. Crest: easily cupped in one hand and easily bent from side to side. | >19–22 |
| 3 |  | Neck: enlarged and thickened. Crest: palpable from the poll to the withers, filling a cupped hand, and beginning to form longitudinal fat deposits on both sides of the neck. | >22–27 |
| 4 |  | Neck: very enlarged and thickened. Crest: grossly thickened with fat deposits from the poll to the withers, forming longitudinal bands of fat on both sides of the neck. Crest cannot be bent easily from side to side. | >27–34 |
| 5 |  | Neck: very enlarged and thickened. Crest: very thickened with hard fat deposits, rounded along both sides of the neck. | >34 |

**Table 2** Association of the fatty neck score (FNS) with the morphometric measurements of neck adiposity.

| Morphometric measurements | FNS (no = 53) | |
| --- | --- | --- |
| | $r_s$[a] | $p$[b] |
| 0.25 NC[c] | 0.37 | 0.007 |
| 0.50 NC[e] | 0.42 | 0.002* |
| 0.75 NC[c] | 0.40 | 0.003* |
| Mean NC[d] | 0.44 | 0.001* |
| 0.50 NC:NL | 0.35 | 0.011 |
| Mean NC:NL | −0.01 | 0.925 |
| 0.50 NC:NH[e] | −0.58 | <0.001* |
| Mean NC:NH | −0.83 | <0.001* |
| 0.50 NH | 0.83 | <0.001* |
| 0.50 NC:NT[f] | −0.68 | <0.001* |
| Mean NC:NT | −0.82 | <0.001* |
| 0.50 NT | 0.83 | <0.001* |

Notes.
[a] Spearman rank correlation coefficient.
[b] $P$ value.
[c] Neck circumference (NC) at 0.25–0.50–0.75 of the neck lenght (NL).
[d] Average of 0.25 NC, 0.50 NC, 0.75 NC.
[e] Neck height (NH).
[f] Neck thickness (NT).
*Bonferrroni-corrected statistically significant.

to perform in the field were included (FNS, BCS, NH, NT and dental score). The lactation month and age were also included since they are physiological features of the animals.

The inter-observer reliability of the expert evaluator and owners in their assessment of BCS and FNS was evaluated by means of intra-class correlations, and by means of Kendall's Coefficient of concordance.

# RESULTS

## Population description

Fifty-three healthy lactating donkeys of various breeds, including 7 Martina Franca, 10 Ragusano, 2 Romagnolo and 34 crossbreeds, with a median age of 9 years (Interquartile range: 7–12 years), an estimated median body weight (BW) of 314.5 kg (Interquartile range: 269–350) and a mean month of lactation of 4 ± 3 months, were evaluated.

## Fatty neck score definition and its association with morphometric measurement of the neck

Among the considered morphometric parameters, FNS was found to be positively correlated to 0.50 NH and 0.50 NT ($p < 0.001$), 0.50 NC ($p = 0.002$), 0.75 NC ($p = 0.003$), mean NC ($p < 0.001$) and negatively correlated to the mean NC:NH and mean NC:NT, 0.50 NC:NT and 0.50 NC:NH ratios ($p < 0.001$) (Table 2).

**Table 3  Association of the body condition score (BCS) with the morphometric measurements of the body.**

| Morphometric measurements | Researchers' BCS (no = 53) | |
|---|---|---|
| | $r_s$[a] | $p$[b] |
| BW[c] | 0.42 | 0.11 |
| HG[d] | 0.16 | 0.27 |
| AC[e] | 0.25 | 0.07 |
| SH[f] | −0.01 | 0.95 |
| OP[g] | 0.05 | 0.70 |
| FNS[h] | 0.84 | <0.001[*] |
| AC:BW | 0.01 | 0.96 |
| AC:HG | 0.21 | 0.13 |
| AC:OP | 0.24 | 0.08 |
| AC:SH | 0.35 | 0.01 |
| HG:BW | −0.10 | 0.49 |
| HG:SH | 0.14 | 0.31 |
| HG:OP | 0.06 | 0.69 |

Notes.
[a] Spearman rank correlation coefficient.
[b] $p$ values.
[c] Body weight.
[d] Heart girth.
[e] Abdominal circunference.
[f] Shoulder-hip lenght.
[g] Olecranon tuber-pinbone lenght.
[h] Fatty neck score.
[*] Bonferrroni-corrected statistically significant.

## Association of BCS with morphometric measurements and dental score

The median BCS and FNS of the lactating donkeys were 2.5 (2–3) and 2.5 (1.5–3), respectively. No significant correlation was found between the morphometric measurements and BCS (Table 3). However, our results highlighted a positive and significant correlation between BCS and FNS ($p < 0.001$; Table 3).

Kendall's coefficient of concordance between the expert evaluators' and the owners' scores for the BCS was low (0.28), thus indicating a substantial disagreement in their evaluations. The owner gave higher score to the animals, result that was clearly at odds with the evaluations of the expert evaluators, who were trained in BCS scoring; it was found that donkeys with a larger abdominal circumference received a higher BCS since a significant, but also rather low correlation was found between AC and the owners 'estimations of BCS ($r_s = 0.41$, $p = 0.002$).

The intra-class correlation coefficients were calculated to estimate the reliability of the scores (BCS, FNS), when assigned by expert evaluators. The assigned scores showed an intra-class correlation coefficient of 0.85 for BCS (95% CI [0.80–0.92]) and of 0.58 (95% CI [0.30–0.76]) for FNS.

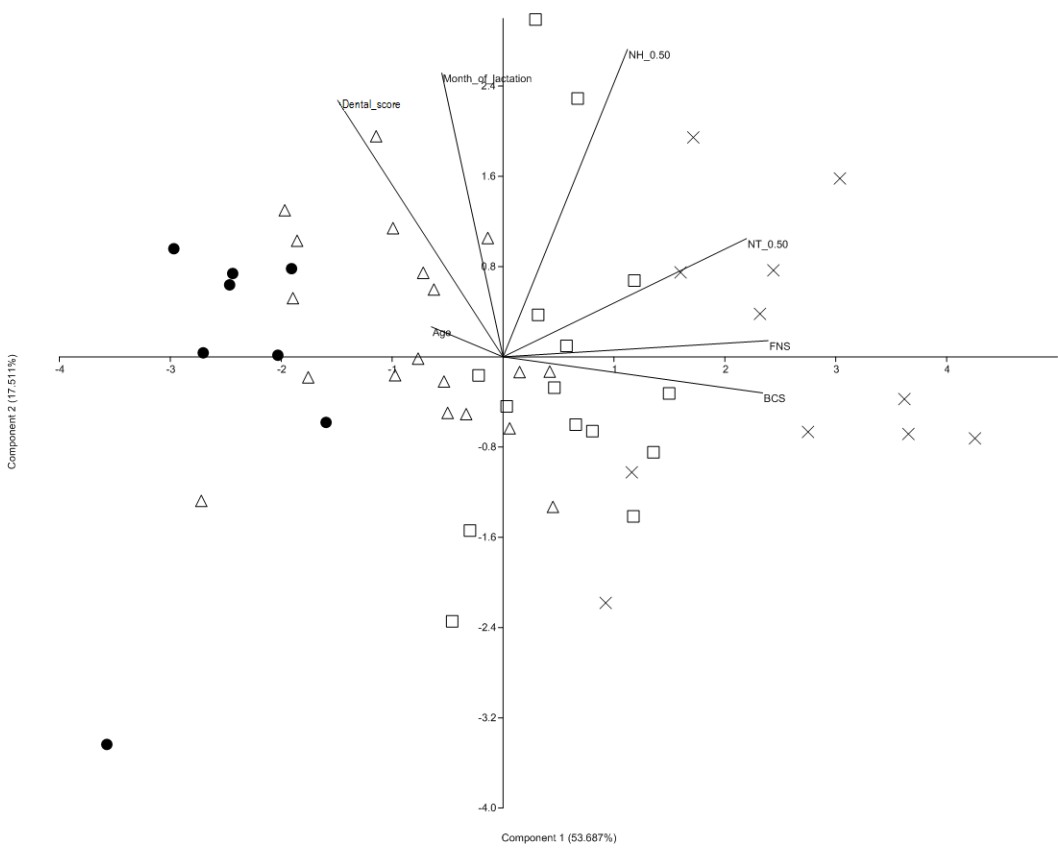

**Figure 3  Principal component analysis performed on selected animal-based indicators of the body condition. In order to improve the visualization of the PCA results, different symbols were assigned arbitrarily.** ● body condition score (BCS) < 2; △ BCS < 3; □ BCS < 4; x BCS > 4. NT, neck thickness; FNS, fatty neck score; NH, neck height.

A Chi-squared test ($p < 0.05$) indicated a significant relationship between BCS and the dental score. The donkeys that had a BCS score of 1 and 2 showed the highest proportion of score 2 (poor dental conditions), when assessed on the basis of the dental score.

## PCA analysis of animal-based indicators of the body condition

A principal component analysis (PCA) was performed in order to represent the variability of the selected animal-based indicators (BCS, FNS, dental score, NT, NH, age and month of lactation) of the 53 lactating donkeys. The suitability of the data for PCA was evaluated (KMO = 0.80; Barlett's test, $p < 0.001$). Figure 3 shows that PCA separated the classes of BCS on the first principal component (PC1): component 1 explains 54% of variance of the data, and component 2 another 18%, for a total of 72% of variability. PC1 was positively correlated to FNS, BCS, NT and to NH, with high positive loadings. Conversely, the presence of dental abnormalities showed high negative loadings on PC1, thus indicating that the animals with high BCS had high FNS, as well as poor dental conditions. Table 4 shows the loadings of the variables of the first and second principal components, and shows how each variable contributes to each component. The subjects belonging to a specific

**Table 4  PCA loadings of selected variables for the monitored lactating donkey population.**

| | Components | |
| --- | --- | --- |
| | 1 (53.69%) | 2 (17.51%) |
| Age | −0.224 | 0.498 |
| Month of lactation | −0.208 | 0.758 |
| BCS[a] | 0.896 | −0.045 |
| NT[b] 0.50 | 0.915 | 0.291 |
| NH[c] 0.50 | 0.883 | 0.316 |
| FNS[d] | 0.944 | 0.042 |
| Mouth condition | −0.594 | 0.463 |

Notes.

[a] Body condition score.

[b] Neck thickness at 0.50 of neck lenght.

[c] Neck heigh at 0.50 of neck lenght.

[b] Fatty neck score.

breed were not grouped separately or as outliers for BCS in the principal component analysis graph.

## DISCUSSION

The body condition is generally estimated by considering various types of morphological or physiological measurements (*Labocha & Hayes, 2012*). The most common animal-based indicator used to assess the body condition of donkeys is the body condition score (BCS), which is proposed as an index of the overall adiposity. In fact, the BCS system includes appraisal, both visually and by means of palpation, of the adipose tissue, which is then scored either on a 5-point or a 9-point scale (*Pearson & Ouassat, 2000*; *Burden, 2012*). Correlations between the BCS, as an index of the overall adiposity, and animal-based morphometric measurements have already been demonstrated in horses (*Carter et al., 2009*; *Dugdale et al., 2011*). However, in the present study, when the proposed animal-based morphometric indicators of the body were analysed, no correlation was found between them and the BCS. This has led the authors to question the suitability of the morphometric measurements used in the study as indicators of the overall adiposity of this species. This lack of correlation could be due to several factors: when employing BCS to measure the overall adiposity of the animal, it is necessary to bear in mind that there is a certain level of subjectivity in the assignment of scores (*Carter et al., 2009*) and, according to *Dugdale et al. (2011)*, there is a loss of sensitivity of subjective BCS systems in overweight subjects. Another factor is related to the fact that donkeys are not small horses, although they both belong to the Equidae family; donkeys also differ from each other in many ways, particularly as far as their anatomical variations and physical conformations are concerned (*Burden & Thiemann, 2015*). This variability is thus not only interspecific, it is also intraspecific. In addition, *Kugler, Grunenfelder & Broxham (2008)* stated, on the basis of an overview of the current donkey population in Europe, that most animals are crossbreeds and cannot be categorized into specific breeds. The donkeys in the present study were also mainly crosses, unlike most other livestock species, in which pedigree-breeding and high genetic selections usually exist.

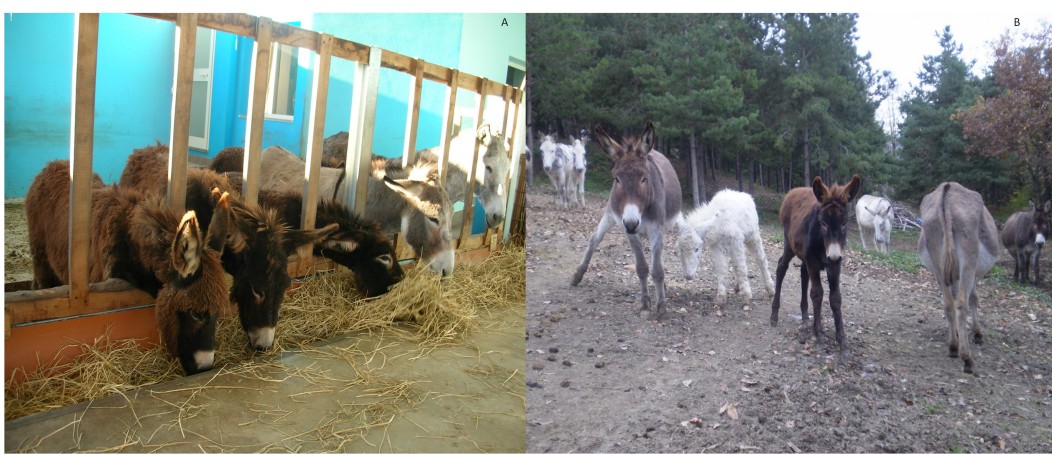

**Figure 4** **Diversity of the donkey population in Italian breeding farms.** Lactating jennies and their foals at the feeding trough (A) and in a woody open field (B).

Cross breeding in the donkey population has resulted in diversity amongst individuals, in particular, for example in body size (Fig. 4). Therefore, morphometric measurements, in spite of being suitable for horses and ponies and although they are easy to perform, cannot be considered an objective alternative to the evaluation of the body condition of lactating donkeys. Although no correlation was found between the morphometric measurements and BCS, when evaluated by the expert evaluators, producers usually rely on morphometric measurements to evaluate BCS. Interestingly, when the donkey owners were asked to evaluate the BCS, they gave higher score to the animals result that was clearly at odds with the evaluations of the expert evaluators, who were trained in BCS scoring; it was also found that donkeys with a larger abdominal circumference received a higher BCS and this can be explained by the fact that the owners were probably misled by the innate anatomical conformation of the abdomen. In fact, donkeys are anatomically characterized by a pendulous abdomen (*Burden, 2012*).

A new, animal-based morphometric indicator of the neck has been analysed in the present study. Regional fat adiposity could be an important indicator that has not yet been included in body condition evaluations. To this end, a new scoring system, based on the assessment of the morphology of the fat deposition, and which has been called Fatty Neck Score (FNS), has been proposed. This new scoring system is based on an evaluation of neck thickness (NT). This adipose tissue of donkeys, unlike that of other equids, tends to droop on both sides of the crest of the neck (*Burden, 2012*). Unlike previous studies conducted on horses, the FNS was not positively correlated to the 0.50 NC:NH, the mean NC:NH, the 0.50 NC:NT, or the mean NC:NT ratios. Instead, FNS was significantly and negatively correlated to these ratios. This result can be explained by considering that the shape of the neck of a donkey is different from that of a horse. The shorter neck and the more protruding manubrium of the donkey support a heavy skull (*Burden & Thiemann, 2015*), and this leads to the development of a remarkably thick *cutaneus colli* muscle, which even covers the middle one-third of the jugular furrow (*Burnham, 2002*). In the present study, it has been possible to develop an objective scale of reference for FNS in a population of

lactating donkeys (Table 1). FNS is a morphometric index of regional fat deposition but even if correlated to BCS, it can be independent of the overall adiposity status, since it can remain even when the overall body weight decreases (*Burden, 2012*; *Burden & Thiemann, 2015*). In addition, this regional adiposity could play a different role in donkeys from the role it plays in horses. It is well known that the Cresty Neck Score in horses and ponies (*Carter et al., 2009*; *Giles et al., 2015*) can be linked not only to the body condition, but also to the metabolic status, but this aspect has not yet been characterized in donkeys, and further studies are necessary to understand its metabolic purpose.

The mechanism for the overall determination of the body condition is too complex to be explained only through a correlation and univariate analysis. PCA was therefore used to indicate the most useful components to define the body condition of lactating donkeys. It was performed on animal-based indicators which can be performed easily on animals (FNS, NH, NT) and that showed a significant correlation to BCS. Age and month of lactation were also included. PC1 displayed high loadings for the FNS, BCS and NH and FNS was the main variable that contributed to PC1, thus suggesting that it is important for the description of the body condition. The results obtained in the present study make it possible to speculate that FNS could be a useful farm animal-based indicator, in addition to BCS, in defining the body condition of donkeys. Nevertheless, further studies are needed to investigate whether there is a link between FNS and the hormonal status of donkeys and disease. Furthermore, the findings of the present study suggest that dental disorders should be included when evaluating the body condition of lactating donkeys. According to *Rodrigues et al. (2013)*, dental disorders, such as sharp points and hooks, are recognized as major, but often unnoticed and therefore often untreated, disorders of equids. In addition, several studies have demonstrated that dental disorders in donkeys are associated with poor BCS, weight loss (*Du Toit et al., 2008*; *Du Toit, Burden & Dixon, 2009*) and colic (*Cox et al., 2007*). These results are supported by the present results, which indicate a significant inverse relationship between BCS and oral conditions.

## CONCLUSIONS

A new scoring system, called Fatty Neck Score (FNS), has been proposed in the present study. The results underline the fact that measurement of body condition is complex task. According to PCA, it is necessary to evaluate FNS, BCS and furthermore the present study suggest that the dental condition of donkeys should also be considered as a farm-based indicator. However, in order to evaluate which animal-based indicators can improve the accuracy of the evaluation of the body condition of donkeys, more studies are required, even to understand how training can help to avoid misjudgements of the body condition. The authors believe that caregivers might use BCS, FNS and dental scores together as a tool for the evaluation of the body condition of donkeys, as long as they are trained in accurate evaluation.The present results apply to those animals that are specifically bred for milk production.

### Funding

The present study was supported by the grant of the University of Turin (ex 60% Linea B, 2013). The funders had no role in study design, data collection and analysis, decision to publish, or preparation of the manuscript.

### Grant Disclosures

The following grant information was disclosed by the authors:
University of Turin.

### Competing Interests

The authors declare there are no competing interests.

### Author Contributions

- Emanuela Valle conceived and designed the experiments, performed the experiments, analyzed the data, contributed reagents/materials/analysis tools, wrote the paper, prepared figures and/or tables, reviewed drafts of the paper.
- Federica Raspa performed the experiments, wrote the paper, reviewed drafts of the paper.
- Marzia Giribaldi analyzed the data, wrote the paper, reviewed drafts of the paper.
- Raffaella Barbero performed the experiments, contributed reagents/materials/analysis tools, reviewed drafts of the paper.
- Stefania Bergagna and Sara Antoniazzi contributed reagents/materials/analysis tools, reviewed drafts of the paper.
- Amy K. Mc Lean performed the experiments, reviewed drafts of the paper.
- Michela Minero analyzed the data, reviewed drafts of the paper.
- Laura Cavallarin conceived and designed the experiments, performed the experiments, wrote the paper, reviewed drafts of the paper.

### Animal Ethics

The following information was supplied relating to ethical approvals (i.e., approving body and any reference numbers):

Regional surveillance working group, Protocollo 9641/DB2017, 23/03/12.

### Field Study Permissions

The following information was supplied relating to field study approvals (i.e., approving body and any reference numbers):

This paper describes the results of a surveillance program put in place by health authorities of the Region (regional surveillance working group, Protocollo 9641/DB2017, 23/03/12). All the procedures of the study that do not involve any invasive experimental work were part of the activities of the regional surveillance working group and were conducted in the presence of regional veterinary services. The group was working for the production of the regional guidelines for donkey milk production (Regione Piemonte, BU29 18/07/2013 Codice DB2017 D.D. 17 giugno 2013, n. 461).

## Data Availability
The raw data has been supplied as Data S1.

## Supplemental Information
Supplemental information for this article can be found online at http://dx.doi.org/10.7717/peerj.3001#supplemental-information.

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

## FURTHER READING

**Bruynsteen L, Moons CPH, Janssens GPJ, Harris PA, Vandevelde K, Lefère L, Duchateau L, Hesta M. 2015.** Level of energy restriction alters body condition score and morphometric profile in obese Shetland ponies. *Veterinary Journal* **206**:61–66 DOI 10.1016/j.tvjl.2015.06.006.

**Pearson RA, Ouassat M. 1996.** Estimation of the liveweight and body condition of working donkeys in Morocco. *The Veterinary Record* **138**:229–233 DOI 10.1136/vr.138.10.229.

**Pleasant RS, Suagee JK, Thatcher CD, Elvinger F, Geor RJ. 2013.** Adiposity, plasma insulin, leptin, lipids, and oxidative stress in mature light breed horses. *Journal of Veterinary Internal Medicine* **27**:576–582 DOI 10.1111/jvim.12056.