# Peer review of "A functional approach to the body condition assessment of lactating donkeys as a tool for welfare evaluation"

_PeerJ, doi:10.7717/peerj.3001_

## Round 0.1 · original submission · Major Revisions

Please state your aims clearly, and give more details about the statistical analysis that you performed, specially in relation to the sample size.

One important point is that your discussion includes a paragraph about differences between individuals and between breeds, and this was not accounted for in the analysis.

I would appreciate your consideration of all reviewers suggestions (and note the two annotated PDFs supplied - the journal staff will send you a Word doc for the PDF from Reviewer 1).

·

Basic reporting

Insufficient clarity. Aims not very clear.
Sentences overlong and overpunctuated.
Codes are provided for some variables, but not all, and uniformity would be preferable. There are too many opportunities for a reader to become confused.

Experimental design

Seems good enough. Is reproducible.

Validity of the findings

Very reliant on statistical procedures, and it is not too clear how appropriate and robust these are. Their nature and purpose should be better described.
Authors are careful to specify that the results apply to a small sample, but do not really spell out how this sample could be different to the general population of donkeys.

Additional comments

Because I earn my living as a professional editor, I found it impossible to read through this paper WITHOUT editing it - for language, but also by providing comments and questions. The paper's authors are welcome to see any or all of these. I usually work in Word, using Track Changes, but had difficulty with conversions. I shall therefore attempt to send the resulting Word version which shows my alterations and comments.

Reviewer 2 ·

Basic reporting

Please see specific comments on the pdf.

General comments:
There are areas of the ms that need clarification.
In particular:
1) state the aims clearly.
2) Indicate why this ms focuses on lactating donkeys in particular.
3) Clarify the terms BCS, adiposity, FNS, morphometric measurements. The relationshop between these measurements and how they are distinct from one another was not always clear.
4) In some areas the text is long and could be more concise.

Experimental design

One concern is that a number of different breeds of donkey were included in the analysis, The discussion includes a paragraph about differences between individuals and between breeds, yet this was not accounted for in the analysis. Explanation of why not would be useful.

Validity of the findings

Please see comments on pdf

Additional comments

Please see comments on pdf

Annotated reviews are not available for download in order to protect the identity of reviewers who chose to remain anonymous.

·

Basic reporting

No comments

Experimental design

No Comments

Validity of the findings

No Comments

Additional comments

This paper is an orginal research subject and proposes a pratcical new method for donkey farmers to easily evaluate body condition.
Concerning english writting, I noticed some small errors on lines 29, 56, 66, 77, 79, 108.
In gerneral, too much spaces between words to be corrected.
Titles and legends for tables and figures are not homogeneous. To be reviewed.

---

## Round 0.2 · Minor Revisions

I would like that you take into account these last suggestions of our reviewer (see attached PDF).

Reviewer 2 ·

Basic reporting

The writing has improved considerably. There are still some areas that are not clear and where the wording could be improved. Also, some suggestions made in my previous review have not be addressed completely. I have made suggestions on the pdf attached.

Literature and ms structure are fine.

Experimental design

Research question is clearer, although the aims/objectives could still be more succinct. Suggestions on the pdf.

Validity of the findings

Data appears robust, and limitations have been included. I am not familiar enough with PCA to be able to assess that part of the statistical analysis.

Additional comments

The writing has improved and most of my comments have been addressed.
However, again I have made a number of additional suggestions in the ms that I think will help.
I still recommend asking a native English speaker to read the final version - this would help the flow and clarity.

Annotated reviews are not available for download in order to protect the identity of reviewers who chose to remain anonymous.

---

## Round 0.3 · accepted · Accept

I am delighted that your manuscript is ready for publication. Thank you for following the suggestions of our reviewers.